# Histological Identification and Quantification of Eosinophils and Ascites in Leghorn Chickens Treated with High Oral Concentrations of NaCl–Pilot Study

**DOI:** 10.3390/toxics10070381

**Published:** 2022-07-09

**Authors:** Victor M. Petrone-Garcia, Inkar Alejandro Castellanos-Huerta, Saeed El-Ashram, Marco A. Juárez-Estrada, Benjamin Fuente-Martínez, Danielle B. Graham, Guillermo Tellez-Isaias

**Affiliations:** 1Livestock Sciences Department, Facultad de Estudios Superiores Cuautitlán, Universidad Nacional Autónoma de México, Cuautitlán Izcalli 54714, Mexico; vmpetrone@hotmail.com; 2Facultad de Medicina Veterinaria y Zootecnia, Universidad Nacional Autónoma de México, Ciudad Universitaria, Ciudad de Mexico 04510, Mexico; britoco@unam.mx (M.A.J.-E.); benjaminfuente@yahoo.com.mx (B.F.-M.); 3College of Life Science and Engineering, Foshan University, Foshan 528231, China; saeed_elashram@yahoo.com; 4Department of Poultry Science, Division of Agriculture, University of Arkansas, Fayetteville, AR 72701, USA; bmahaffe@uark.edu (D.B.G.); gtellez@uark.edu (G.T.-I.)

**Keywords:** basophils, ascites, demyelination, renal tubulonecrosis, hepatosis, NaCl poisoning, poultry, salt

## Abstract

The purpose of this pilot study was to determine the role played by eosinophils in NaCl poisoning and right cardiac hypertrophy (ascitic syndrome) in Leghorn chickens, as well as the histological findings in the central nervous system (CNS), liver, and kidney. Moreover, the hypertrophy of the right ventricle index (HRVI) as an indicator of ascites was evaluated. Male SPF Leghorn birds at 28 days of age were submitted to two experiments. Food and water (FW) experiment: birds were treated with food plus 3.3% NaCl for the next 27 days and 1% NaCl in their drinking water from days 22 to 27. Water experiment (W): birds were treated with 1% NaCl in their drinking water for 5 days. In both experiments, the chickens exhibited loss of appetite, diuresis, and watery, green diarrhea during treatment days; at 24–27 td-FW and experiment W, the birds showed nervous signology (prostration, running movements, tremors, and comatose state). In the leukogram at 28 td-FW, an increase (*p* < 0.05) in heterophiles and basophils was observed. CNS eosinophilia was not observed in birds intoxicated with NaCl, though they did present demyelination in the brain and spinal cord, hepatic degeneration, mesangial proliferative glomerulopathy, and acute proximal renotubular necrosis.

## 1. Introduction

NaCl poisoning has been observed in cattle, sheep, pigs [1,2], rabbits [3], birds [4], and humans [5,6]. The NaCl in feed can be toxic to young chicks and poults. However, the most toxicity results from consuming saline water, with or without water deprivation. The excess NaCl in the feed or water causes significant economic losses in poultry in many countries [4].

The clinical signs reported in birds and mammals have been mainly related to the type of intoxication. The acute type usually manifests as excessive NaCl consumption, and accidental and nervous signs are observed, including blindness, seizures, deafness, drowsiness, comatose state, and death [7]. Chickens chronically intoxicated with salt also exhibit digestive disorders such as diarrhea, anorexia, dehydration, polydipsia, and death [8]. The macroscopic lesions that have been reported include myocardial hemorrhage, hydropericardium, dilation, and right ventricular hypertrophy [7].

Birds have three kinds of granular leukocytes (granulocytes)—heterophiles, eosinophils, and basophils—in the circulatory system and mast cells in the tissues. Each avian granulocyte contains more than one type of granule, and there are differences between bird species. Both heterophiles and eosinophils have eosinophilic cytoplasmic granules, so they are called acidophiles when they cannot be distinguished in tissues through conventional stains. To differentiate the two acidophiles, histochemical techniques should be used, since eosinophils, unlike heterophiles, are peroxidase positive [9]. The staining for p-phenylenediamine dihydrochloride plus pyrocatechol (PPD + PC) specifically marks the peroxidase of eosinophil granules and not of heterophile granules [10]. Basophils are much more numerous in birds than in mammals, their granules are deeply basophilic, and their nucleus is unilobular and pale. Ziehl–Neelsen (ZN) staining specifically identifies basophils and mast cells [9,11].

Histological findings at the brain level in cases of NaCl encephalopathy [12] are necrosis, congestion, edema, and the presence of eosinophilic infiltrations, although reports about the presence of eosinophils are contradictory, since some studies, such as those conducted by Wages [13] and Mohanty [14], reported eosinophils in turkeys and chickens, respectively, and Baze [12] did not observe them in chicks. Osweiler et al. [15] mentioned eosinophilia in the brain as a characteristic finding in cases of NaCl poisoning in birds and pigs.

Regarding the findings in the kidney, Puette and Crowell [16], measuring the glomerular diameter, found that one of the incipient changes of renal injury was the increase in the glomerular diameter and the proliferation of mesangial cells. Siller [17], in his review of renal pathology in birds, points out the low importance given to the variety of lesions that have been reported by different researchers in cases of NaCl poisoning, such as acute tubular necrosis (nephrosis), glomerulopathy, and nephrosclerosis. Necrosis has been reported in the liver of birds poisoned with NaCl [7], as have congestion and edema [13].

In pigs, NaCl poisoning produces eosinophilic meningoencephalitis [2,18]. In birds, there are contradictory findings. Trainer and Karstad [19] observed eosinophilic encephalitis in natural NaCl poisoning but could not replicate it experimentally. Mohanty and West [14] induced eosinophilic myelomeningitis in 15% of chickens experimentally intoxicated with NaCl. Baze [12] reported no polymorphonuclear infiltration in chickens that were intoxicated with NaCl and treated with corticosteroids or a diuretic substance (ethacrynic acid). Wages et al. [13], after natural intoxication in turkeys, found encephalitis with perivascular infiltration of acidophiles, but because they did not use special stains, no differentiation between heterophiles and eosinophils was made.

The increase in the number of erythrocytes and the increase in blood flow resistance have been suggested as causes of increased pulmonary artery pressure, and both have been attributed to sodium intoxication [4,20]. High levels of NaCl can produce rigidity and increase the number of erythrocytes [9]. Both factors increase blood flow resistance. On the other hand, ascites syndrome has been characterized by increased pulmonary artery pressure and hypertrophy of the right ventricle, of which NaCl poisoning has been mentioned as a possible cause [7,8,21]. Julian [8] evaluated the cardiac hypertrophy index of broilers poisoned with NaCl. First, they weighed the birds, and then they removed the heart to eliminate atria, fat, and major vessels, carefully separating the right ventricle, including the left valve. They weighed the septum, then weighed the right ventricle, and, to obtain the total weight, added the left ventricle and the septum. They evaluated the relationships between ventricular and body weights as well as ventricular and total weights. They observed early development of pulmonary hypertension syndrome (ascites syndrome) in those birds that were given high levels of NaCl in the diet.

Due to the inconsistencies in the literature on the role played by eosinophils and the little information on histological lesions in birds that consumed high concentrations of salt, it is necessary to deepen the study of avian eosinophils in the diagnosis of NaCl intoxication and to have more elements to evaluate the inclusion levels of salt in poultry nutrition. Furthermore, hypertrophy of the right ventricular index (HRVI) has not been evaluated in Leghorn birds that consumed high concentrations of salt; therefore, it is relevant to investigate Leghorn birds subject to NaCl intoxication.

The objective of this research was to determine the role played by eosinophils in NaCl poisoning and right cardiac hypertrophy (ascitic syndrome) in Leghorn chickens, as well as the histological findings in the central nervous system (CNS), liver, and kidney in specific pathogen–free (SPF) Leghorn chickens treated with NaCl in their food and drinking water from day 28 to day 54 of age.

## 2. Materials and Methods

### 2.1. Animals for Experimentation

The study was conducted according to the guidelines of the Declaration of Helsinki and approved by Internal Committee for Care and Use of Experimental Animals (CICUAE) of the National Autonomous University of Mexico (UNAM). Ethical approval code: CICUAE-FESC C20_06.

A total of 450 28-day-old SPF male Leghorn birds were housed in conventional laying cages in the same isolation units. Chickens were fed ad libitum with standard starter feed for laying hens (Appendix A).

### 2.2. Preparation of Food and Drinking Water with NaCl

− Food with 3.3% NaCl: To the commercial feed (J. Baker Laboratory, Edo. Mex.), 3% NaCl was added to obtain a total final concentration of 3.3%.− Drinking water with 1% NaCl: NaCl (1%) was added to the drinking water.

### 2.3. Design of the Two Experiments

The chickens were fed with NaCl in amounts ten times higher than the requirements to evaluate the clinical signs and lesions of non-fatal chronic intoxication, but above all, to evaluate the role of eosinophils and basophils. Likewise, 1% was supplied in drinking water, which is a lethal dose, with which the terminal injuries of intoxication would be found. The treatment schedule was selected considering the pullets no longer needed heaters and weekly feeding phases [22].

Food and water experiment (FW): Intoxication with 3.3% NaCl in food and 1% NaCl in drinking water. Two groups of 180 chickens were used. Chickens in group FW1 were fed with food with 3.3% NaCl for 27 days, and they drank drinking water in a 1:2 ratio (feed:water) for the first 21 days, followed by water with 1% NaCl from day 22 through day 27. Chickens in group FW2 (control group) consumed feed without extra salt plus drinking water in a 1:2 ratio (feed:water) for the whole 27 days (Table 1).

Water experiment (W): Intoxication with 1% NaCl in drinking water. Two groups of 45 chickens were used. Group W1 was given drinking water with 1% NaCl. Chickens in group W2 (control group) were given drinking water ad libitum.

### 2.4. Obtaining the Samples

In both experiments, the birds that showed ascites or died were counted. The birds were euthanized utilizing carbon dioxide (CO_2_) inhalation [23] when indicated by the sampling schedule (Table 1), or the humane endpoint approach was considered when the birds were in agony and presented nervous signs such as prostration with tremors, running movements, or comatose state.

FW experiment: Samples were taken on treatment days (td) 7, 14, 21, and 27 or on the day that the birds presented nervous signs, which occurred for 24 birds of group 1 between 24 and 27 td-FW. W experiment: Samples were taken when the chicken presented nervous signs, which occurred between 3 and 5 td-W. The birds of the FW2 and W2 groups were sampled on the same treatment days as were the FW1 and W1 groups (Table 1).

### 2.5. Histology

Tissue collection. Liver: A 20-mm-length × 15-mm-width × 2-mm-thick cross-section of the middle part of the left hepatic lobe was obtained. Kidney: A 2-mm-thick cross-section of the right kidney’s total surface of the middle part of the cranial lobe was made. Brain: The brain was sectioned longitudinally to observe the median-sagittal fully plane of the prosencephalon and rhombencephalon, and the spinal cord was cut between the fifth and sixth cervical vertebrae, obtaining 3-mm-thick portions for histological examination to observe the total cross-sectional area of the cut. The portions of the organs were kept in 10% buffered formalin (3.75% pure formaldehyde concentration), embedded in paraffin, cut into 4-µm-thick sections, and stained with hematoxylin–eosin (HE) [24].

### 2.6. Identification of Granulocytes

The sample was embedded in paraffin, sectioned to 4 µm, and counterstained with Harris hematoxylin. The basophils were differentiated with ZN staining [24], PPD + PC staining was used to differentiate eosinophils from heterophiles. To do the PPD + PC staining, the fixed organs were incubated in a substrate composed of 10 mg PPD (SIGMA Chemical Co., St. Louis, MO, USA), 20 mg PC (Mallinckrodt. Baker, Phillipsburg, NJ, USA), and 10 mL Tris buffer (0.1 M) at a pH of 7.6. The incubation lasted 180 min, with a substrate refreshing at 90 min [25].

### 2.7. Leukogram

FW experiment: Four samples were taken from the left cutaneous ulnar vein of the birds at 7, 14, 21, and 27 days of the experiment or when the birds began to show nervous signs. W experiment: Sampling was performed when the birds began to show nervous signs. Blood was taken from each bird, and 0.9 mL of blood with 0.1 mL of 10% EDTA was evaluated. The smears were examined with Wright’s stain [24].

### 2.8. Evaluation of the Tissue Samples

#### Histological Evaluation

Under the immersion objective (100×) of light microscopy, heterophiles, basophils, and eosinophils were identified from the total surface of the sections stained with HE, PPD + PC, and ZN, respectively. With the 40× objective the rest of the lesions were evaluated in HE-stained slides.

Liver. The rows of leukocytes surrounding the portal spaces and central veins were counted and given the following score: 0 = no granulocytes or lymphocytes were present; 1 = 1 to 4 rows of granulocytes or lymphocytes; 2 = 5 to 6 rows of granulocytes or lymphocytes; 3 = more than 7 rows of granulocytes or lymphocytes. To evaluate hepatic degeneration, the loss of cellular contour or the presence of vacuoles in the cytoplasm of the hepatocyte was considered: 0 = neither loss of contour nor vacuoles were observed in hepatocytes; 1 = 1% to 10% of the liver parenchyma was affected; 2 = 10% to 20% of the liver parenchyma was affected; 3 = more than 20% of the liver parenchyma was affected.

Kidney. Twenty-five glomeruli were observed randomly in each section to evaluate the degree of mesangioproliferative glomerulopathy according to the percentage occupied by the glomerulus out of the whole glomerular space: 0 = none observed; 1 = the glomerulus occupied the vascular pole or up to 25% of the glomerular space. 2 = it occupied 26 to 50% of the glomerular space; 3 = it occupied 51 to 75% of the glomerular space; 4 = it occupied 76 to 100% of the glomerular space. Twenty-five fields of each section were observed randomly to evaluate acute tubular necrosis (nephrosis) by counting the number of cells in necrosis and distinguishing whether the necrotic cells were in proximal or distal tubules.

Brain and spinal cord. A granulocyte count was performed, and the degree of demyelination was determined according to the percentage of the affected parenchyma. Demyelination was considered when the axons were dilated and contained granules: 0 = demyelination was not observed; 1 = demyelination in less than 10% of the parenchyma; 2 = demyelination in 10 to 20% of the parenchyma; 3 = demyelination in more than 20% of the parenchyma.

### 2.9. Leukogram Evaluation

We counted 100 cells that had differentiated into heterophiles, eosinophils, basophils, lymphocytes, and monocytes, yielding the percentage of each cell type. To obtain their absolute values, the total leukocyte count was multiplied by each percentage of differentiated cells. Total and relative leukocyte counts were performed as described by Campbell [26].

### 2.10. Hypertrophy of the Right Ventricle Index (HRVI)

For dead and euthanized chickens of both groups, the HRVI was calculated. The HRVI was calculated as the ratio of the weight of the right ventricle to the weight of the left ventricle, including the interventricular septum. The weight was exact up to 1/100 g.

### 2.11. Statistical Analysis

The number of samples per variable group was 45. Data for the leukocytes count are presented as mean with standard deviation. The normality of their distribution was confirmed by the Shapiro–Wilk test, and their homoscedasticity was verified by Levene’s test. Accordingly, the parametric Student’s t test was performed. The HRVI was subjected to a square root arcsine transformation before analysis. Scores of acute proximal tubular necrosis, mesangioproliferative glomerulopathy, brain and spinal cord demyelination, and percentage of degenerated hepatocytes are presented as the median in a boxplot. Although the hypotheses of normal distribution (Shapiro–Wilk test) and homoscedasticity (Levene test) were confirmed for the scores for acute proximal tubular necrosis, mesangioproliferative glomerulopathy and percentage of degenerated hepatocytes, it was considered that the scores are discrete values; therefore, the non-parametric two-tailed Mann–Whitney U test was performed. The Mann–Whitney U test was also performed for the brain and spinal cord demyelination scores since the hypotheses of normal distribution and homoscedasticity were not confirmed. For ascites and chicken mortality, the χ2 test was applied. The statistical significance was set at *p* < 0.05.

## 3. Results

FW experiment. Chickens exhibited loss of appetite, diuresis, and watery, green diarrhea during the 27 days of treatment. Only after they drank water with 1% NaCl did the birds show nervous signs such as prostration, running movements, tremors, comatose state, and death. As soon as we noticed that the birds were comatose, we euthanized them: three at 24 td-FW, five at 25 td-FW, 10 at 26 td-FW, and three at 27 td-FW. No ascites or hypertrophy of the right ventricle was observed in any of the samples. Only one chicken, at 21 td-FW, presented mild hydropericardium.

W experiment. Chickens exhibited loss of appetite, diuresis, and watery, green diarrhea during treatment. At 3 td-W (nine birds), 4 td-W (12 birds), and 5 td- W (24 birds) the nervous signs were observed. No chicken had hydropericardium or ascites.

Histology. In the kidneys of the groups treated with NaCl (Figure 1), greater (*p* < 0.05) degeneration and acute necrosis of proximal tubules (Figure 2a) and mesangioproliferative glomerulopathy (Figure 2b) were observed at 24–27 td-FW and at all times in the W experiment than in the control groups (Figure 2c).

The brains of the groups treated with NaCl showed greater (*p* < 0.05) demyelination (Figure 2d) at 24–27 td-FW in the FW1 group and at all times in the W1 group than those of the FW2 and W2 groups, respectively (Figure 3). In the spinal cord, greater (*p* < 0.05) demyelination (Figure 2e) was observed at 21 td-FW, 24–27 td-FW, and 3–5 td-W (Figure 3).

In the liver (Figure 4), greater (*p* < 0.05) proteinaceous (Figure 2g), vacuolar degeneration and hyperemia (Figure 2h) were found in birds at 24–27 td-FW and 3–5 td-W than the control groups (Figure 2i). The rest of the birds of both experiments did not show differences (*p* > 0.05). No tissue infiltration by eosinophils, basophils, or heterophiles was seen in any birds.

Leukogram. At 27 td-FW, an increase (*p* < 0.05) in heterophiles and basophils was observed in the group treated with NaCl (Table 2). No differences were observed (*p* > 0.05) in differential leukocyte count from the rest of the sampling times.

Hypertrophy of the right ventricle index (HRVI). In the FW experiment, the group treated with 3.3% NaCl had an HRVI of 0.22 ± 0.043, while the group treated with 0.3% NaCl had an HRVI of 0.21 ± 0.039 (*p* > 0.05).

## 4. Discussion

The birds that were administered 1% NaCl in drinking water (W experiment) and the birds at 24–27 td-FW presented mortality and more severe lesions than the birds at 7, 14, and 21 td-FW, agreeing with Julian et al. [8], who mentioned that sodium in drinking water can cause greater effects because birds consume 1.5 to 2.5 times more water than food, and sodium, whether added to water or food, increases water consumption. However, blood sodium or chloride levels were not evaluated in this investigation to confirm which of these two elements or their interaction cause injury and mortality due to high levels of NaCl in feed or drinking water.

The lesions found in the kidney and liver are similar to those observed by Mohanty and West [14], Martindale [27], and Sokkar et al. [28], who observed that the most affected tubules were the proximal tubules. The presence of mesangioproliferative glomerulopathy agrees with the reports of Puette et al. [16] and Nishimura [29].

Demyelinating lesions were observed in the brain and spinal cord, as in the study by Baze [12], but granulocytes were not present, unlike in Wages et al. [13], who reported them in naturally NaCl-intoxicated turkeys, and Mohanty and West [14], who observed them in an experimental intoxication with NaCl in chickens, so we can conclude that the presence of granulocytes is not a characteristic lesion of NaCl intoxication in birds. Likewise, the presence of lymphocytes could not be related in this case to the intoxication with NaCl because there were no significant differences in lymphocytes between groups. Chickens intoxicated with NaCl did not present granulocytic infiltrate in the CNS, which is consistent with Baze [12], who did not observe granulocytic infiltration in chickens poisoned with 1 or 1.5% NaCl in their drinking water. Additionally, Sinovek et al. [30] found no eosinophilia in quail poisoned with 0.6, 1.0, 2.0, or 3.0% NaCl in the food. Our findings disagree with those of Mohanty and West [14], Trainer and Karstad [19], and Wages et al. [13], who observed granulocytic infiltrate in the CNS in chickens, pigeons, and turkeys intoxicated with NaCl. This discrepancy could be due to the lack of specific stains to identify eosinophils in their investigations and the fact that eosinophils are heterophiles present in an infection secondary to toxicosis.

The findings of demyelination in the brain agree with the signology and with the form of administration, since the birds at 24–27 td-FW and the birds of the W experiment showed nerve signology, and in both cases, 1% NaCl was administered in the drinking water.

The degeneration of hepatocytes observed in this work is consistent with that reported by Islam et al., who described degeneration, necrosis changes and hemorrhage [7].

In the leukogram, we observed significant differences in the counts of heterophiles and basophils at 27 days after starting the treatment between treated groups and the control group. These chickens, in addition to having the chronic intake of NaCl in the feed and the drinking water, were the chickens whose histology presented more severe lesions, and whose blood heterophiles increased as a reaction to intake of high amounts of NaCl. In humans, neutrophils increase in number under NaCl poisoning [5]; the increased neutrophils have been linked to degenerative [31] or toxic processes [32]—processes found in this research.

Regarding the findings of the HRVI index, we observed that the excess of NaCl in the diet of Leghorn birds does not cause an increase in HRVI and ascites syndrome. However, it has been reported that excess NaCl in broilers causes an increase in pulmonary pressure [8] and the rigidity of erythrocytes [21], which could trigger the ascitic syndrome. In Leghorn chickens, the described causes did not cause ascites. The lack of ascites might have been due to genetic factors related to the growth rate since Huchzermeyer [33], comparing different strains of broilers, found a difference in the presence of ascites syndrome, and Lubritz et al. [34] mentioned different results of cardiac weights related to heritability.

## 5. Conclusions

In conclusion, this investigation found that ingesting high amounts of NaCl in food or drinking water does not produce blood eosinophilia nor eosinophilic infiltrate in the central nervous system (CNS), liver, and kidneys. These high concentrations also do not produce ascites syndrome or increased HRVI. However, they produce degenerative and necrotic lesions in the CNS, liver, and kidneys.

## Figures and Tables

**Figure 1 toxics-10-00381-f001:**
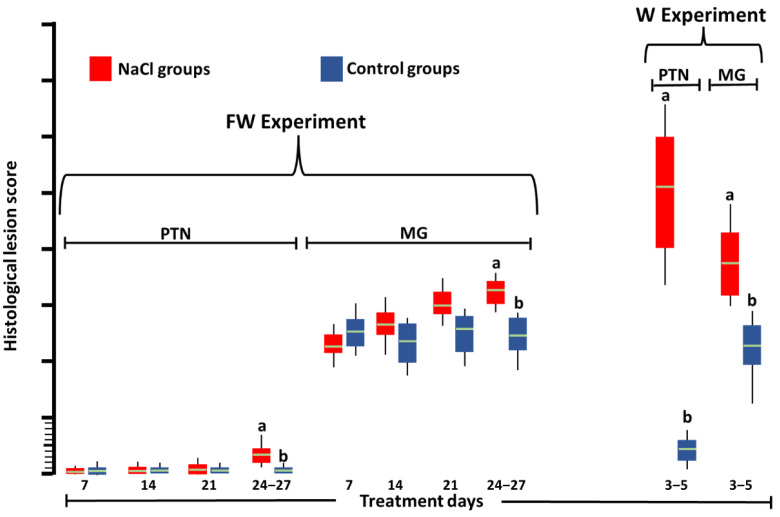
Scores of acute proximal tubular necrosis (PTN) and mesangioproliferative glomerulopathy (MG) in Leghorn SPF chickens at 27 days of treatment with 3.3% NaCl in feed and del 22–27 days 3.3% NaCl in drinking water (FW experiment) or at 5 days of 1% NaCl in drinking water (W experiment). Different letters above boxes in the same experiment and lesion denote significant differences (*p* < 0.05). Green line: median.

**Figure 2 toxics-10-00381-f002:**
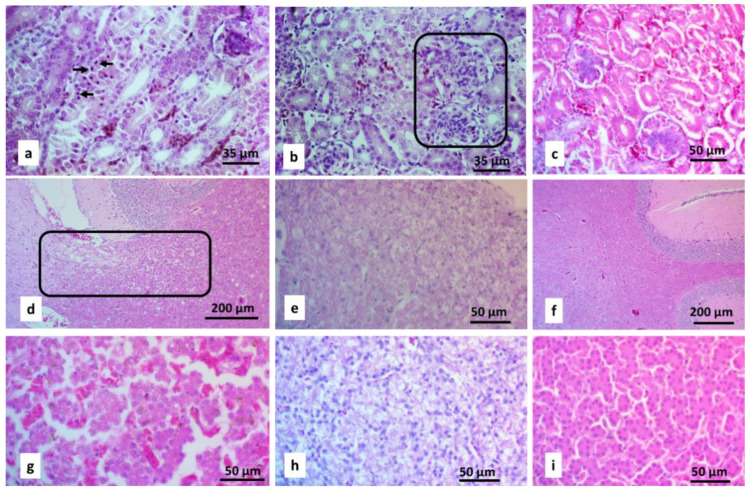
Micrographs of HE-stained organs from chickens treated with 1% NaCl in the drinking water. (**a**) Kidney: pyknotic nuclei characteristic of acute proximal tubular necrosis (arrows). (**b**) Kidney: mesangioproliferative glomerulopathy (inset). (**c**) Kidney from the control group that consumed food with 0.3% NaCl and drinking water. (**d**) Brain with white matter demyelination (inset). (**e**) Spinal cord with white matter demyelination. (**f**) Brain from a control chicken that consumed food with 0.3% NaCl and drinking water. (**g**) Liver: thin granules degeneration and loss of hepatocyte contours (proteinaceous degeneration). (**h**) Liver: thick vacuolar (hydropic) degeneration and (**g**,**h**) dilatation of blood vessels (hyperemia) are observed. (**i**) Liver from a control chicken that consumed food with 0.3% NaCl and drinking water.

**Figure 3 toxics-10-00381-f003:**
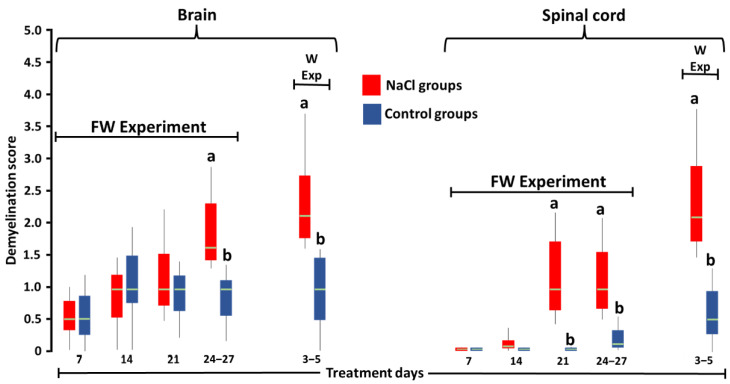
Score of axon demyelination of the brain and spinal cord neurons of SPF Leghorn chickens at 27 days of treatment with 3.3% NaCl in food and 5 days with 1% NaCl in drinking water (FW experiment) or at 5 days of treatment with 1% NaCl in drinking water (W experiment). Different letters above boxes for the same experiment and day denote significant differences (*p* < 0.05). Green line: median.

**Figure 4 toxics-10-00381-f004:**
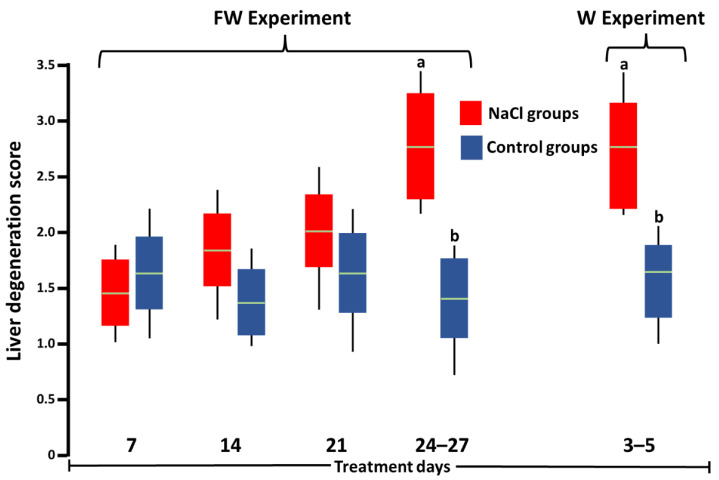
Score of the percentage of degenerated hepatocytes of SPF Leghorn chickens at 27 days of treatment with 3.3% NaCl in feed and 5 days with 1% NaCl in drinking water (FW experiment) or at 5 days of treatment with 1% NaCl in drinking water (W experiment). Different letters above boxes for the same experiment denote significant differences (*p* < 0.05). Green line: median.

**Table 1 toxics-10-00381-t001:** Number of birds, treatments, treatment day (td) and treatment sampling times Leghorn chickens.

			Sampling Time						
			(Treatment Day)						
Experiment	Group	Treatment	3	4	5	7	14	21	24–27
FW	1	NaCl (food, water) ^1^	-	-	-	45 ^n^	45	45	21 ^2^
	2 (control group)	Drinking water and 0.3% NaCl in food ^3^	-	-	-	45	45	45	21
W	1	1% NaCl in water ^4^	9 ^5^	12	24	-	-	-	-
	2 (control group)	Drinking water and 0.3% NaCl in food ^3^	9	12	24	-	-	-	-

^1^ Birds in the FW1 group consumed feed containing 3.3% NaCl at 27 td, and 1% NaCl was added to the same birds in the drinking water from 22 to 27 td. ^2^ Twenty-four birds died, so samples were only taken from 21 birds, which already showed nervous signs between day 24 and day 27 of treatment. ^3^ The birds consumed feed with 0.3% NaCl and drinking water. ^4^ The birds drank water with 1% NaCl from days 1 to 5 of treatment. ^5^ Samples were taken from birds that showed nervous signs. ^n^ Number of birds.

**Table 2 toxics-10-00381-t002:** Differential leukocyte count of SPF Leghorn birds at 27 days of treatment (td) with 3.3% NaCl in feed and 7 days with 1% NaCl in drinking water (FW experiment).

Groups	Total	Lymphocytes	Monocytes	Heterophiles	Eosinophils	Basophils
FW experiment (27 days td)	12522.08 ± 9192.40 (100%)	6003.77 ± 5201.13 (47.95%)	1000.21 ± 686.64 (7.99%)	5029.37 ± 3266.48 (40.16%) ^a^	255.11 ± 353.86 (2.04%)	233.62 ± 198.19 (1.87%) ^a^
Control	7892.19 ± 4428.99 (100%)	5060.03 ± 2603.80 (64.11%)	596.97 ± 363.44 (7.56%)	2138.13 ± 1197.72 (27.09%) ^b^	45.76 ± 51.93 (0.58%)	51.3 ± 88.11 (0.65%) ^b^

Data expressed as mean absolute leukocyte count ± standard deviation (% = mean relative leukocyte count). ^ab^ Different superscripts within columns indicate a significant difference at *p* < 0.05. The rest of the treatment days did not show differences between the two groups (*p* > 0.05).

## Data Availability

Not applicable.

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
