# Peer review of "Histological Identification and Quantification of Eosinophils and Ascites in Leghorn Chickens Treated with High Oral Concentrations of NaCl–Pilot Study"

_toxics, 2022, doi:10.3390/toxics10070381_

Round 1
Reviewer 1 Report
The introduction seems too long. I propose to edit the introduction to make it more concise and legible
The objective of the study at the end of introduction should be reedited to show a novelty of the study.
A precise justification for the selection of NaCl doses must be added. Why NaCl has been added on day 28 to day 54 of age, and not for example from day 20 or 30 to day 50 or 60. It is also must be explained.
Novelty of the study must be underlined in discussion. Results of present study should be analyzed more carefully compared to the previous studies. What innovative results (compared to previous studies) have been observed during the present study.
Conclusion section must be reedited. In present form only shortened results, not conclusion are presented in this section
Author Response
We are very grateful for the time spent on the review and your invaluable corrections and comments.
The introduction seems too long. I propose to edit the introduction to make it more concise and legible.
Intro reduced by 30%
The objective of the study at the end of introduction should be reedited to show a novelty of the study.
Objectives completely reedited
A precise justification for the selection of NaCl doses must be added. Why NaCl has been added on day 28 to day 54 of age, and not for example from day 20 or 30 to day 50 or 60. It is also must be explained.
Justifications made (Lines
Novelty of the study must be underlined in discussion. Results of present study should be analyzed more carefully compared to the previous studies. What innovative results (compared to previous studies) have been observed during the present study.
The discussion has corrections; however, there are no more articles on the topic to include
Conclusion section must be reedited. In present form only shortened results, not conclusion are presented in this section
Conclusion completely reedited
Reviewer 2 Report
Description of methods and results of histology and haemathology requires higher quality.

Author Response
Please see the attachment
Review: Manuscript ID: toxics-1776970
We are very grateful for the time spent on the review and your invaluable corrections and comments.
General: The aim of the study is not clearly explained. The authors mentioned experiment from day 28 to 54 of age, but I do not understand why the 1% NaCl in drinking water was used at the beginning of the experiment, in different age of chickens, not in comparison with the same age of chickens treated with 3.3% NaCl in food and next 1% NaCl in water. Could you explained this discrepancy? And add more description to second experiment (W) – age and breed (Leghorn?) – from text it is not clear.
Objectives were totally restructured
The treatment in drinking water was carried out at the beginning and at the end of the ka those of the birds. Please read lines 134-139. Thank you.
Also the obtaining of samples is not written properly. The schema of experiment needs to be prepare before its start, not at the base of clinical signs as authors added in text: Samples were taken when the chicken presented nervous sings such as tremors, running movements, or comatose state, which occurred between 3 and 5 td-W. What was the reason for this arrangement of samplings?
We strongly agree that the sampling should be done according to the experiment protocol. However, in this case, according to our endpoint policy, the birds had to be euthanized when they presented agony represented by the comatose state and other clinical signs.
Moreover in Materials and Methods are missing some informations. Examination of blood is not clearly described. Determination of total count of leukocytes is not mentioned. Please, add the data to be possible the control of calculation of absolute white blood cell populations from the relative percentages which are obtained during differentiation of white blood cell on the smears.
Thanks for the timely comments. We add the references that were used for the determination of the leukogram. Total leukocyte count and relative values were also added to results (Table 2).
In Discussion comparison of liver data are missing, and reformulation of some sentences is required.
The sentence referring to liver and other discussions were reformulated
In Literatura follow the instructions of journal for preparing the write format of citations.
We follow the instructions for authors; however, the publisher-sanctioned app had to be set to manually to fix the flaws in the references.
Abstract
Line 27 Please, change the sentence: We also evaluated ...in next: Done
Moreover, the hypertrophy of the right ventricle index (HRV) as an indicator of ascites was evaluated. Changed
Line 29 to one of two experiment – not all chickens was in the same age and breed? Changed sentence
Line 37 Please, correct acute proximal renotubular into: acute proximal renotubular necrosis Changed
Line 39: Please, correct tubulnecrosis into tubulonecrosis. Changed
Introduction
Why you have two empty rows 61 and 62, as well only one 106 in text? I think that it is no needed because you continue with next text. Empty rows removed
Line 137 Please, omit the word count and correct: eosinophil, basophil and heterophils counts,… Changed
Materials and Methods
Line 154 and 156 Please, avoid spacers at the beginning of sentences. Empty rows removed
Line 185 Table 2 is not include. But I think that Table 1 presented the mentioned data. Changed
Line 196 Identification of granulocytes on histological slides? I suppose that it was done not at the smears but on histological slides. Whole incised samples was stained and next embedded in paraffin? It is needed to mention at the beginning of paragraph. Corrected. The sample was embedded in paraffin, sectioned to 4 µm
Line 197 Please, correct the sentence: The basophils were differentiated from the other two types of
granulocytes… Sentence corrected
Line 198 remove one of [24]. Removed
Paragraph Leukogram: How the blood was taken, from which vena, and which staining was used for smears. Add the information, please. Added. Lines 189 and 190
Paragraph Leukogram evaluation mentioned total leukocyte count but the authors not describe the way of determination of total leukocyte. Add the information, please. Information added on lines 193 and 227
Line 261 The sentence mentioned: The scores of demyelination of the brain and spinal cord are presented…but in the obtaining of samples only taking of spinal cord is described, not the area of sampling from brain. Add this information, please. Information added on lines 174-175
Results
Line 277-278 Please, correct the sentence as next: At 3 td-W (nine birds), 4 td-W (12 birds), and 5 td- W (24 birds) the nervous signs was observed. Changed sentence. Lines 258-259
Figure 2 liver g and h) I disagree with the description: liver: thin, gouty liver degeneration and loss of hepatocyte contours (albumin degeneration). First of all gouty means presence of uric acid, and I cannot see crystals. Also if there is albuminous (not albumine) or by other words parenchymatous degeneration (cloudy swelling, granulous degeneration are other synonyms) and its higher degree vacuolar degeneration (in your case h) the size of liver is enlarged and pale. Your picture named g showing interstitial edema the cause of which really may be high amount of albumins, but this need to be prove by biochemistry in blood or by histochemistry. Please, consider rename albumin degeneration into proteinaceous and add in h) vacuolar (synonyme is hydropic) degeneration. Because from the picture is not clear if erythrocytes are present, add if the dilatation of blood vessels (hyperemia) was observed. All corrections were made
Line 331 Please, substitute albumin with cloudy swelling or proteinaceous (if was not prove albumin by special staining). Other possibility is presence of amyloid (Congo red staining) and then amyloid degeneration is the right name of diagnosis. Changed term. Line 278
Line 342-344 Paragraph Leukogram. Blood leukocyte count in all cases (text and Figure) needs the change into white blood cell count or differential leucocyte count. And mention of total leukocytes count is acquired. Changed term. Lines 326, and 328 (Table 2)
Discussion
Line 385-388 The first sentence is mentioned degeneration of hepatocytes, but next text described demyelination included before in lines 369-373. I think that moving of these lesions is needed as well as more information and comparison about the damage liver by NaCl is required. All corrections were made. Lines 362-367
Line 390 Please, correct in the sentence: …between intoxicated and the control groups. Sentence corrected. Lines 369-370
Line 393-395 Please add the citation to this sentence and explain how the neutrophilia may influence degenerative reaction in the tissues, because it is no clear from text. Citation added. Line 373-375
Line 396-398 Please, reformulate the sentence and omit we can conclude, because it is not your own idea. Sentence removed
Line 405-406 Please, reformulate the sentence resulted from misunderstanding what not observed in your study, if polycytemia or ascites. Sentence reformulated. Lines 385-389
Conclusion
Line 412 Please correct: ….and acute tubular necrosis of renal proximal tubules. Sentence removed. Lines 385-389
Literature
Unify citation in doi – by the prescription of journal add them in all papers or remove existing. Remove existing
Citation 7. Can J Vet Res – add the whole name instead of indexes. Add full name
Citation 8. Am J Vet Res – add the whole name instead of indexes. Add full name
Citation 17. J VET Diagn Invest – add the whole name instead of indexes. Add full name
Citation 20. Avian Pathol – add the whole name instead of indexes. Add full name
Citation 22. J Am Vet Med Assoc – add the whole name instead of indexes. Add full name
Citation 24. Histochem J – add the whole name instead of indexes. Add full name
Citation 29. Onderstepoort J Vet Res – add the whole name instead of indexes. Add full name
Reviewer 3 Report
The paper is aimed to describe the effects of an excessive NaCl intake on young chickens. However, in my opinion, there are several important flaws that reduce the scientific soundness of the achieved results and their consequent interpretation.
Despite the Authors repeatedly collected blood, hematic markers were not considered, as well as electrolytes concentration in blood. This seriously impairs the experimental design, as there is no correlation between the sodium concentration and the observed effects. In my opinion, such an association could be crucial, especially because the Authors aimed at solving the inconsistency among the studies already present in the literature. As they are, data are far from definitive, since the procedure and the examined factors did not differ significantly from those previously carried out.
In general, Materials and Methods should be presented more clearly.
Additionally, In my opinion, statistics should be completely revised as discrete values were treated as continuous, and therefore results should be reconsidered accordingly.
Following there are specific comments.
Lines 42-45: Those sentences have been pasted exactly from the cited reference. Please avoid such practice.
Lines 63-70: Please provide references for such statements.
Lines 96-105: It is not clear why granulocytes have been treated separately from the other leukocytes, whose involvement had been presented in the previous subsection.
Line 148: Why SPF chicks were used? Which specific pathogen was excluded?
Lines 154 and 156: It is not clear if NaCl percentages represent the initial or the final concentration.
Lines 161: “fed with food” instead of “fed food”.
Table 1 and in the main text: At line 150 the Authors wrote “Day 28 of the life of the birds was considered day 1 of the experiment”. It is not clear which count the days indicated in the table and on lines 159-164 are referred to. On line 168, in the figure legend, it is indicated that “treatment day 1 = day 28 of life”, and in lines 178-185 the abbreviation “td” is introduced. All those looks correct, but very confusing. I would suggest better clarifying the timeline.
Lines 178-182: Were animals sacrificed? If yes, how?
Lines 179 and 182: Please specify what samples were collected.
Lines 206-209: It is not clear why different criteria were used for the blood collection and the following leukogram.
Line 227: How were glomeruli chosen? Randomly?
Lines 256-266: As for my little statistical knowledge, scores cannot be considered as continuous, but discrete variables. Therefore, it is not correct to use mean, standard deviation and Student’s t test.
Line 271: Please describe, in the material and Method section, if a Humane End Point approach was considered and, if yes, which criteria were adopted.
Lines 360-364: As it is, those statements are speculative.
Lines 364-384: Please see the general comments.
Line 33: Toxemia has not been considered in the present study.
Lines 403-406: This statement is completely speculative, as no data have been presenting to support it.
Author Response
We are very grateful for the time spent on the review and your invaluable corrections and feedback.
The paper is aimed to describe the effects of an excessive NaCl intake on young chickens. However, in my opinion, there are several important flaws that reduce the scientific soundness of the achieved results and their consequent interpretation.
We strongly try to improve the scientific robustness of the results obtained
Despite the Authors repeatedly collected blood, hematic markers were not considered, as well as electrolytes concentration in blood. This seriously impairs the experimental design, as there is no correlation between the sodium concentration and the observed effects. In my opinion, such an association could be crucial, especially because the Authors aimed at solving the inconsistency among the studies already present in the literature. As they are, data are far from definitive, since the procedure and the examined factors did not differ significantly from those previously carried out.
This work is a pilot study of larger investigations that are in progress. Therefore, your suggestions will be taken into account for future work. The title was modified, indicating that the work is a pilot study.
In general, Materials and Methods should be presented more clearly.
All requested corrections in materials and methods have been made.
Additionally, In my opinion, statistics should be completely revised as discrete values were treated as continuous, and therefore results should be reconsidered accordingly.
The statistics were modified considering discrete values, and the results and figures were also modified.
Following there are specific comments.
Lines 42-45: Those sentences have been pasted exactly from the cited reference. Please avoid such practice.
Changed sentence. Lines 43-46
Lines 63-70: Please provide references for such statements.
Citation added. Line 61
Lines 96-105: It is not clear why granulocytes have been treated separately from the other leukocytes, whose involvement had been presented in the previous subsection.
Sentence removed
Line 148: Why SPF chicks were used? Which specific pathogen was excluded?
The excluded pathogens are congenitally transmitted and cause lesions in the brain, kidney and liver, and immunosuppression resulting in secondary diseases. The pathogens are chicken infectious anaemia virus, avian encephalomyelitis, avian leukosis and inclusion body hepatitis, and bacterial infections such as salmonellosis and mycoplasmosis.
Lines 154 and 156: It is not clear if NaCl percentages represent the initial or the final concentration.
Sentence reformulated. Line 130
Lines 161: “fed with food” instead of “fed food”.
Sentence reformulated. Line 142
Table 1 and in the main text: At line 150 the Authors wrote “Day 28 of the life of the birds was considered day 1 of the experiment”.
Sentence removed
It is not clear which count the days indicated in the table and on lines 159-164 are referred to. On line 168, in the figure legend, it is indicated that “treatment day 1 = day 28 of life”, and in lines 178-185 the abbreviation “td” is introduced. All those looks correct, but very confusing. I would suggest better clarifying the timeline. Sentence reformulated. Lines 149-151, and 164
Lines 178-182: Were animals sacrificed? If yes, how?
Sentences added. Lines 159-163
Lines 179 and 182: Please specify what samples were collected.
Sentence added. Please in lines 171-177
Lines 206-209: It is not clear why different criteria were used for the blood collection and the following leukogram.
Sentence added. Lines 189-191 and 227
Line 227: How were glomeruli chosen? Randomly?
Randomly. Lines 208 and 214
Lines 256-266: As for my little statistical knowledge, scores cannot be considered as continuous, but discrete variables. Therefore, it is not correct to use mean, standard deviation and Student’s t test. Statistical analysis reformulated. Lines 234-247
Line 271: Please describe, in the material and Method section, if a Humane End Point approach was considered and, if yes, which criteria were adopted.
Sentences added. Lines 159-163
Lines 360-364: As it is, those statements are speculative.
Sentence removed
Lines 364-384: Please see the general comments.
Sentence removed or reformulated. Lines 334-345
Line 33: Toxemia has not been considered in the present study.
Sentence removed
Lines 403-406: This statement is completely speculative, as no data have been presenting to support it.
Sentence reformulated. Lines 385-389
Many thanks
Round 2
Reviewer 3 Report
In my opinion, the authors replied to all considerations. It is important to underline that it is a pilot study, as further information would be needed to obtain more general conclusions.
My only remark is about the use of SPF chickens, at line 124. In fact, the Authors replied to my comment but did not add the rationale for their choice in the manuscript.
This manuscript is a resubmission of an earlier submission. The following is a list of the peer review reports and author responses from that submission.
Round 1
Reviewer 1 Report
Dear Authors,
In my opinion, the methodology is confusing, and the authors must clarify:
1) Why did they use doses of 3.3% NaCl in the feed and 1% in the water for the treatment?
2) Why was the water treatment (1%) (FW experiment) only on days 28 to 32 of the experiment?
3) Why did the authors make the evaluations with 3 to 5 days of treatment (therefore, the animals with 30 to 32 days of life) in the W experiment while in the FW experiment the authors made the evaluations with the 7 to 27 days of the experiment (animals with 34 to 54 days of life)?
Other important methodological questions that need to be clarified are
1) Did the experiment last 27 or 28 days? or were samples collected on day 27 and animals continued to be treated until day 28?
2) the authors must adequately describe the control groups of the FW and W experiment. In the figures, the authors use the term control group but do not use it in the methodology;
3) The authors inform that they did the hemogram, but they only describe the results of the leukogram. Do the authors have the blood count data? Why didn't they show?
4) In the histological analyses, how many animals were analyzed? Group 1 and 2 (FW) 156 animals (45+45+45+21)? Group 1, and 2 (W) 45 animals (9+12+24)?
Results
The description of the results is not compatible with the objectives of the study.
The authors report that the objective was to evaluate "...the mortality, eosinophil count, basophil and heterophil counts...in specific pathogen-free (SPF) Leghorn chickens treated with NaCl in their food and drinking water from day 28 to day 54 of age."
However, the authors did not adequately describe mortality, right cardiac hypertrophy, and hematological abnormalities observed throughout the experiment.
Figure 1 - Withdraw, this is not a new result.
Figure 2 - Remove, unnecessary, in my opinion.
Figure 3 - The authors inform, in the figure legend, that the results shown are from day 21 of the FW experiment and from day 5 of the W experiment. However, the figure shows results from other days and periods, how is this possible?
Figure 4 - The figure legend informs that the micrographs shown are of animals treated with 1% NaCl (experiment W) however in the description of the results the authors inform that they are the results of the two experiments. How is this possible?
Figure 5 - Authors should search the literature for the appropriate way of expressing non-parametric data. The way in which the data is presented is not acceptable. The authors state in the figure legend that the results shown are from day 28 of the FW experiment and from day 5 of the W experiment. However, results from other days are shown in the figure.
Figure 6 - The authors state in the legend that the results shown are from day 28 of the FW experiment and from day 5 of the W experiment. However, results from other days are shown in the figure.
Figure 7 - The leukocyte count results only show the data obtained on day 27/28 of the FW treatment and do not show the results of the samples obtained on the 7, 14, 21, and 27 days and still ignore the results of the W experiment.
The authors ignore several results that must have been obtained. The authors must clarify this.
Author Response
1) Why did they use doses of 3.3% NaCl in the feed and 1% in the water for the treatment?
The chickens were fed with NaCl in amounts 10 times higher than the requirements to evaluate the clinical signs and lesions of non-fatal chronic intoxication, but above all to evaluate the role of eosinophils and basophils. And later, 1% was supplied in drinking water, which is a lethal dose; with which the terminal injuries of intoxication would be found.
2) Why was the water treatment (1%) (FW experiment) only on days 28 to 32 of the experiment?
The treatment was only given on those days because the birds died, and there were no birds to continue with the treatment for more days.
3) Why did the authors make the evaluations with 3 to 5 days of treatment (therefore, the animals with 30 to 32 days of life) in the W experiment while in the FW experiment the authors made the evaluations with the 7 to 27 days of the experiment (animals with 34 to 54 days of life)?
Because the water treatment (1%) is deadly; therefore, the duration of the treatment was determined by the day of the presentation of the perimortem clinical signs or by the death of the birds.
Other important methodological questions that need to be clarified are
1) Did the experiment last 27 or 28 days? or were samples collected on day 27 and animals continued to be treated until day 28?
The experiment was planned until day 28; however, the birds survived until day 27. The text has already been corrected to only indicate day 27.
2) the authors must adequately describe the control groups of the FW and W experiment. In the figures, the authors use the term control group but do not use it in the methodology;
The term control group was described in the methodology
3) The authors inform that they did the hemogram, but they only describe the results of the leukogram. Do the authors have the blood count data? Why didn't they show?
The term hemogram was changed to leukogram
4) In the histological analyses, how many animals were analyzed? Group 1 and 2 (FW) 156 animals (45+45+45+21)? Group 1, and 2 (W) 45 animals (9+12+24)?
For histological analyses, 156 birds (experiment FW) and 45 birds (experiment W) were used.
Results
The description of the results is not compatible with the objectives of the study.
The authors report that the objective was to evaluate "...the mortality, eosinophil count, basophil and heterophil counts...in specific pathogen-free (SPF) Leghorn chickens treated with NaCl in their food and drinking water from day 28 to day 54 of age."
However, the authors did not adequately describe mortality, right cardiac hypertrophy, and hematological abnormalities observed throughout the experiment.
Los términos mortalidad y hematological abnormalities fueron eliminados de los objetivos. The right cardiac hypertrophy index (HRVI) se encuentra apropiadamente descrito en materials and methods, results, and el último párrafo de discussion
Figure 1 - Withdraw, this is not a new result.
The figure was removed
Figure 2 - Remove, unnecessary, in my opinion.
The figure was removed
Figure 3 - The authors inform, in the figure legend, that the results shown are from day 21 of the FW experiment and from day 5 of the W experiment. However, the figure shows results from other days and periods, how is this possible?
The figure legend of figure 1 (previously figure 3) was modified
Figure 4 - The figure legend informs that the micrographs shown are of animals treated with 1% NaCl (experiment W) however in the description of the results the authors inform that they are the results of the two experiments. How is this possible?
Lesions were found in treated chickens from both experiments. However, the micrographs are examples taken from a single bird, and the bird is from experiment group W.
Figure 5 - Authors should search the literature for the appropriate way of expressing non-parametric data. The way in which the data is presented is not acceptable. The authors state in the figure legend that the results shown are from day 28 of the FW experiment and from day 5 of the W experiment. However, results from other days are shown in the figure.
The way we use to indicate the mean and median in the same bar graph. In our opinion, the reader can easily understand which is the mean and which is the median. However, we are willing to change the way of expressing the data, if you indicate the appropriate reference.
The figure legend of figure 3 (previously figure 5) was modified
Figure 6 - The authors state in the legend that the results shown are from day 28 of the FW experiment and from day 5 of the W experiment. However, results from other days are shown in the figure.
The figure legend of figure 4 (previously figure 6) was modified
Figure 7 - The leukocyte count results only show the data obtained on day 27/28 of the FW treatment and do not show the results of the samples obtained on the 7, 14, 21, and 27 days and still ignore the results of the W experiment.
The authors ignore several results that must have been obtained. The authors must clarify this.
The figure legend of figure 5 (previously figure 7) was modified. Added the text: The rest of the treatment days did not show differences between the two groups (P>0.05)
Reviewer 2 Report
All elements in the following sections are clearly described. I have no questions or comments to the authors.
Author Response
Thanks for your comments
Reviewer 3 Report
NaCl poisoning has been observed in many species and this article may contribute to research on other species.
Author Response
Thanks for your comments
Round 2
Reviewer 1 Report
Dear authors, in my opinion, the results obtained by you were strongly influenced by the moment of obtaining the samples. The authors report that the samples were obtained when the animals showed clinical signs of intoxication or comatose. How many animals were comatose and how many had clinical signs of prostration, tremor, and running movements? Were samples obtained at the beginning of the signs or were they obtained later? What was the criterion for determining the slaughter time to obtain the samples? These are factors that influence the results.
Regarding the presentation of non-normal data. I recommend expression in the median and interquartile range. See more information on https://www.analystsoft.com/en/products/statplus/content/help/analysis_charts_box_plot_box_whiskey.html#:~:text=A%20box%20plot%20is%20a,thus%20more%20resistant%20to%20the
